# Mitochondrial-nuclear cross-talk in the human brain is modulated by cell type and perturbed in neurodegenerative disease

Aine Fairbrother-Browne[1,2,3], Aminah T. Ali [3], Regina H. Reynolds [2], Sonia Garcia-Ruiz[2,4], David Zhang[2,4], Zhongbo Chen [1,2], Mina Ryten [2,4,5✉] & Alan Hodgkinson [3,5✉]

Mitochondrial dysfunction contributes to the pathogenesis of many neurodegenerative diseases. The mitochondrial genome encodes core respiratory chain proteins, but the vast majority of mitochondrial proteins are nuclear-encoded, making interactions between the two genomes vital for cell function. Here, we examine these relationships by comparing mitochondrial and nuclear gene expression across different regions of the human brain in healthy and disease cohorts. We find strong regional patterns that are modulated by cell-type and reflect functional specialisation. Nuclear genes causally implicated in sporadic Parkinson's and Alzheimer's disease (AD) show much stronger relationships with the mitochondrial genome than expected by chance, and mitochondrial-nuclear relationships are highly perturbed in AD cases, particularly through synaptic and lysosomal pathways, potentially implicating the regulation of energy balance and removal of dysfunction mitochondria in the etiology or progression of the disease. Finally, we present *MitoNuclearCOEXPlorer*, a tool to interrogate key mitochondria-nuclear relationships in multi-dimensional brain data.

[1] Institute of Neurology, University College London (UCL), London, UK. [2] Genetics and Genomic Medicine, Great Ormond Street Institute of Child Health, University College London, London WC1E 6BT, UK. [3] Department of Medical and Molecular Genetics, School of Basic and Medical Biosciences, King's College London, London, UK. [4] NIHR Great Ormond Street Hospital Biomedical Research Centre, University College London, London, UK. [5] These authors contributed equally: Mina Ryten, Alan Hodgkinson. ✉email: mina.ryten@ucl.ac.uk; alan.hodgkinson@kcl.ac.uk

Tissues of the central nervous system (CNS) are not only highly energetically demanding, consuming 20% of the body's total energy supply[1], but heterogeneous in their requirements, with wide variation in energy demands across their constituent cell types[2,3]. As such, matching energy supply to demand is a tightly regulated process and dysfunction in these processes has been linked to a wide range of neurodegenerative diseases (NDs)[4–6]. Energy production in the CNS is largely dependent on mitochondria, which have their own compact genomes that code for proteins of the electron transport chain. However, most of the proteins required for normal mitochondrial function are encoded in the nucleus, making interactions between the two genomes vital for key cellular processes such as mitophagy, calcium buffering, cellular signalling and apoptosis[7]. Transcription of nuclear-encoded mitochondrial proteins occurs in the nucleus and translation is carried out by cytoplasmic ribosomes before the products are imported into mitochondria[8]. Through these processes, the mitochondria are fully resourced to fulfil their numerous and integral roles in the cell.

In neuronal cell types, the mitochondrial−nuclear relationship is particularly complex. Neurons are highly dependent on oxidative phosphorylation (OXPHOS), rendering them vulnerable to oxidative stress induced by reactive oxygen species (ROS). Given that OXPHOS components are bi-genomically encoded, while the components of the 'ROS defence system' (RDS) are nuclear-encoded, coordinated provision of these factors is required to maintain both continuous ATP production and neuronal integrity[9]. Furthermore, given that neurons are terminally differentiated cells, they rely heavily on bi-genomically encoded autophagic pathways for removal of dysfunctional organelles as well as misfolded and aggregated proteins in order to maintain function throughout life[10]. Additionally, neurons have a unique and highly specialised architecture, requiring them to ensure a consistent supply of nuclear-encoded mitochondrial proteins to large quantities of mitochondria, across many metres in some instances[2,11].

Given the intricacy and scale of mitochondrial−nuclear coordination required in human brain tissue, there is ample opportunity for dysfunction. In neurons, failure of coordinated mitochondrial clearance and biosynthesis contributes to disease pathogenesis. This can be seen in the aetiology of Parkinson's disease (PD), where mutations in *PINK1* and *PARK2* are associated with autosomal-recessive PD and their protein products have been implicated not only in mitophagy, but also mitochondrial biogenesis[12,13]. However, pathology of the mitochondrial biogenesis and quality control pathways is not unique to PD. Analyses of brain samples from individuals with Alzheimer's disease (AD) have shown that levels of the mitochondrial biogenesis transcriptional 'master-regulator' PGC-1α in hippocampal tissues are reduced relative to control tissue, suggesting that disruption of PGC-1α-dependent pathways contributes to pathogenesis[14]. Collectively, this evidence points to a role for dysfunction of the mitochondrial−nuclear relationship in NDs.

Despite this, the analysis of mitochondrial−nuclear cross-talk at scale is mostly limited, focusing either on a small number of features, or a small number of samples through the analysis of absolute RNA expression values[15], in vivo work involving single gene knockdown[16], or indirectly analysing mitochondrial function by measuring metabolite output[17]. Larger studies that have looked at cross-talk in multiple tissues include a population-level analysis of expression quantitative trait loci (eQTLs) associated with the expression of mitochondrially encoded genes, and a multi-tissue analysis of nuclear and mitochondrial gene expression correlations[18,19]. These studies support the complexity and functional relevance of mitochondrial−nuclear relationships in the brain but lack CNS-specificity and analysis of potential processes and pathways most relevant to mitochondrial−nuclear coordination.

Here, we focus specifically on mitochondrial−nuclear relationships in CNS tissues using RNA sequencing data from a large number of individuals from multiple cohort studies. We find that across the CNS, there is regional variation in co-expression likely modulated by cell-type-specific processes, reflective of functional specialisation in the brain. We identify disease-specific patterns in mitochondrial−nuclear relationships that are important for understanding the aetiology of neurological disease.

## Results

Since mitochondrial processes are important in brain tissue and their perturbation is thought to have a role in ND, we aimed to identify whether relationships between expression levels of mitochondrial- and nuclear-encoded genes are variable across brain regions, cell types and ND status. To do this, we calculated pairwise Spearman correlation coefficients between all nuclear and mitochondrial gene pairs, after regressing out covariates (see 'Methods'). We leveraged data across 12 CNS tissues from the Genotype-Tissue Expression (GTEx) project for analyses in healthy tissue, and frontal cortex tissue from the Religious Orders Study/ Memory and Aging Project (ROSMAP) AD dataset for analyses in a case−control paradigm.

**Correlations in mitochondrial−nuclear gene expression are variable across the human CNS.** In order to investigate correlations in mitochondrial−nuclear gene expression across all CNS regions, we calculated Spearman correlation coefficients for each pair of nuclear and mitochondrial-encoded genes (15,001 and 13 genes respectively, making a total of 195,013 comparisons) in each of the 12 GTEx CNS regions. Distributions of the correlation values for each CNS region were visualised as density plots to facilitate cross-CNS comparison (Fig. 1a). We observed that CNS regions have distinct and varying mitochondrial−nuclear correlation distributions. While some regions showed Gaussian-like distributions (cerebellar hemisphere, hypothalamus, substantia nigra) (Fig. 1c), others showed dispersed distributions, containing more high magnitude relationships, and fewer neutral correlations (caudate basal ganglia, putamen basal ganglia) (Fig. 1b). Qualitative analysis revealed mitochondrial−nuclear distribution similarity within GTEx CNS tissues derived from the same broad regional classification (fore-brain, mid-brain and hind-brain). We quantitatively confirmed this through unsupervised Euclidean clustering of regional correlation coefficients across all CNS tissues. This identified biologically meaningful clusters, whereby cortical regions and distinct regions of the basal ganglia (putamen, nucleus accumbens and caudate) were grouped together (Supplementary Fig. 1), which appears to reflect functional specialisation in the human brain.

**Regional cell-type composition contributes to distinct regional mitochondrial−nuclear correlation distributions in the CNS.** We hypothesised that regional differences in cell-type composition may be contributing to regional differences in mitochondrial −nuclear correlation profiles. To test this, we considered whether cell-type markers were enriched at the positive and negative extremes of the correlation coefficient distributions. This analysis was performed for each GTEx CNS region using the Expression Weighted Cell-type Enrichment (EWCE) method, which tests whether a given set of genes is expressed more highly in a cell-type of interest than might be expected by chance[20]. Cell-type specificity data were derived from two human brain snRNA-seq experiments, the first of which used middle temporal gyrus nuclei[21], and the second used hippocampus and prefrontal cortex nuclei[22]. The input to this method was nuclear-encoded genes derived from gene pairs in the highest 5% of positive correlations and highest 5% of negative correlations for each region.

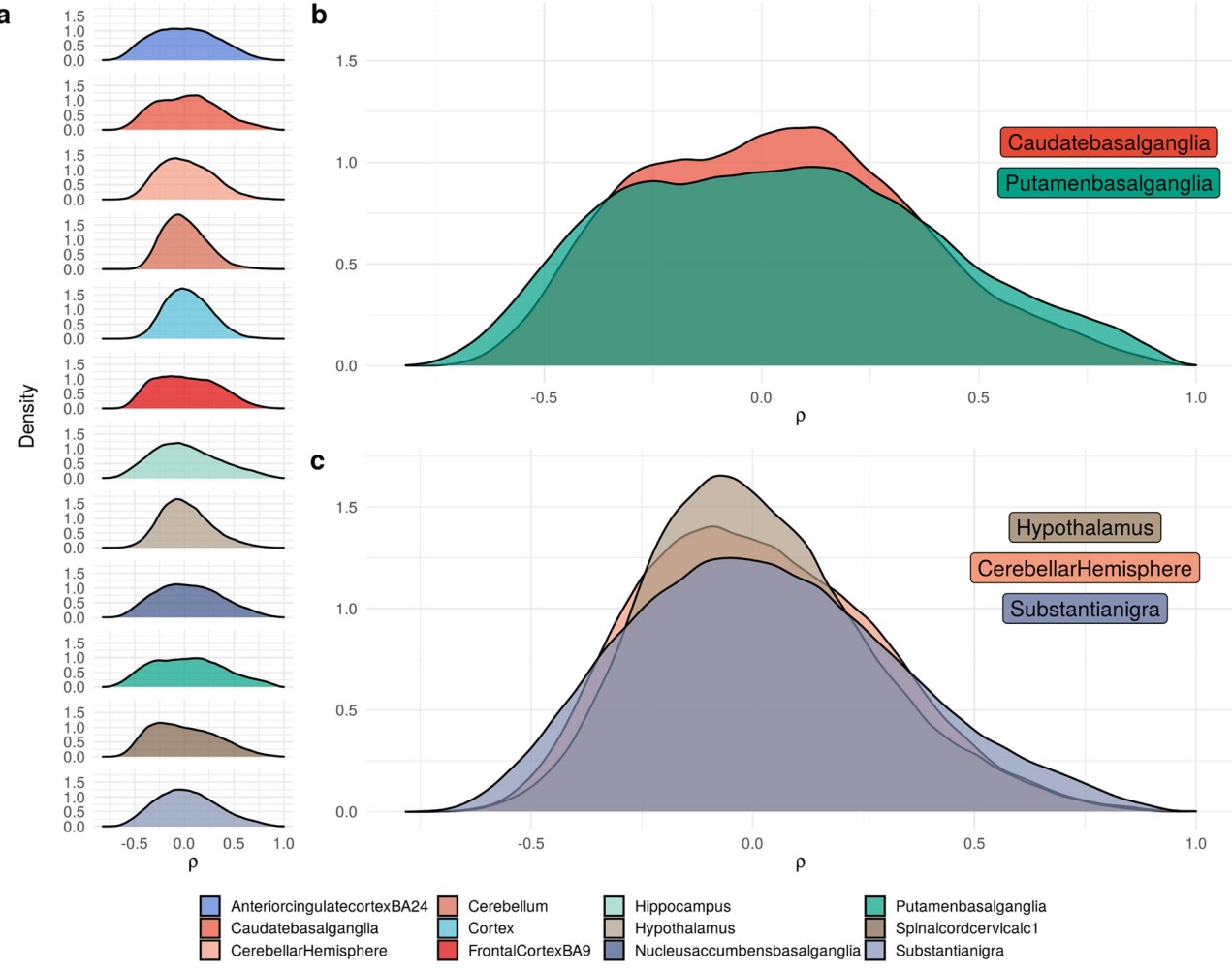

**Fig. 1 Distributions of mitochondrial−nuclear correlation coefficients (ρ) in GTEx CNS regions. a** Mitochondrial−nuclear ρ distributions for 12 GTEx CNS regions. **b** Panel to show ρ distributions of the putamen basal ganglia and caudate basal ganglia regions overlaid. **c** Panel to show distributions of the cerebellar hemisphere, hypothalamus, and substantia nigra regions overlaid.

We found that genes with a high specificity for neuronal cell-types (GABAergic and glutamatergic) were significantly enriched ($P < 0.05$, Bonferroni-corrected for regions and gene sets) in negative mitochondrial−nuclear gene pairs across CNS regions (Fig. 2). In contrast, genes with a high specificity for non-neuronal cell-types (astrocytes, microglia) were significantly enriched in positive mitochondrial−nuclear gene pairs ($P < 0.05$ in 6/12 regions for astrocytes; $P < 0.05$ in 5/12 regions for microglia, Bonferroni-corrected for regions and gene sets), the exception to this being oligodendrocytes (Fig. 2). A strong cross-CNS signal for oligodendrocyte marker enrichment was observed in negatively correlated pairs ($P < 0.05$ in 10/12 regions, Bonferroni-corrected for regions and gene sets), coupled with no significant enrichment detected in positively correlated pairs. For astrocytes and microglia, we observed a trend towards marker enrichment in positive pairs over negative pairs across the CNS. Reassuringly, we note that related regions display similar cell-type enrichment profiles, indicative of biological functionality being reflected in these enrichments. For example, GTEx-defined[23] technical sample replicates (the cortex and frontal cortex, and cerebellum and cerebellar hemisphere) as well as regions closely biologically associated such as the basal ganglia (putamen, nucleus accumbens and caudate), demonstrate consistent patterns of cell-type enrichment.

To further test our hypothesis, we used published cell-type proportion estimates[24] to determine whether correcting GTEx expression data for the effect of cell-type proportions would result in more homogenous cross-CNS mitochondrial−nuclear correlation profiles. To this end, we included five GTEx regions (see 'Methods') for which we determined the cell-type proportions to be most representative (Supplementary Fig. 2A, B), and compared the distributions of cross-regional Spearman correlation variances per mitochondrial−nuclear gene pair with and without correction for cell-type proportions. Applying this approach, we find that the distributions of variances are significantly different to each other (two-sample Wilcoxon signed rank test, $P < 2.2e−16$), but the medians of both distributions are also significantly higher than 0 (one-sample Wilcoxon signed rank test, $P < 2.2e−16$ for mitochondrial−nuclear distributions derived from both correction strategies) (Supplementary Fig. 2C, D). As such, we conclude that cell-type proportion is a modulator of cross-CNS variation in mitochondrial−nuclear correlations, but note that regional specialisations still exist after correcting for cell-type proportions.

**Post-synaptic processes are enriched in mitochondrial−nuclear gene pairs that are highly variable across the CNS.** Having established the importance of cell-type composition in driving variation in mitochondrial−nuclear correlation profiles in the CNS, we aimed to identify biological processes associated with this

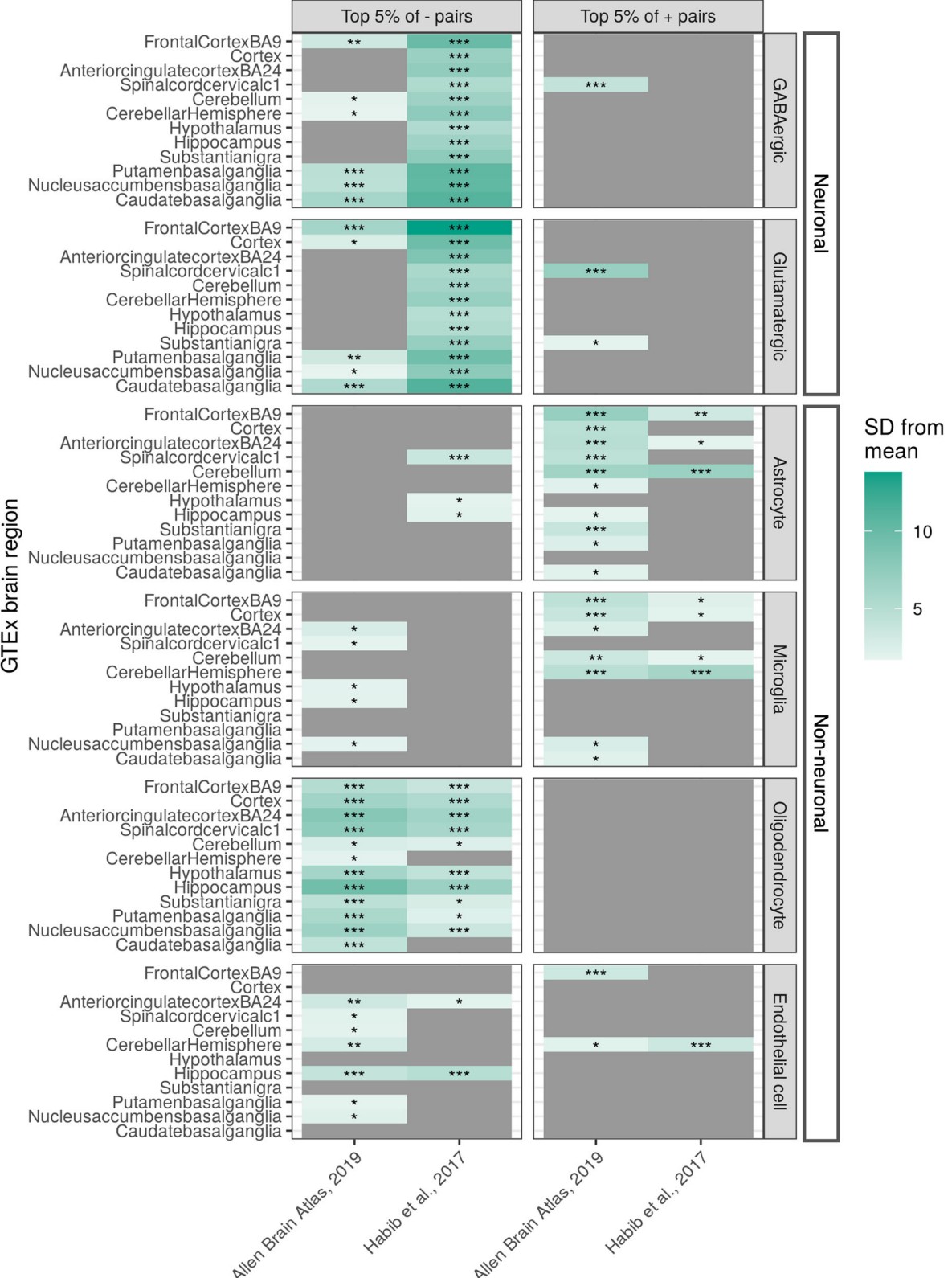

**Fig. 2 EWCE-derived cell-type enrichments for 12 GTEx CNS regions.** The left-hand *y* axis refers to the GTEx CNS region, while the right-hand facet labelling refers to the cell-type. For each cell-type in each region, the metric for enrichment is shown as the number of standard deviations from the bootstrapped mean (SD from mean, indicated by the colour bar). A dark green colour indicates a higher number of standard deviations from the mean, while lighter green indicates fewer standard deviations from the mean. The *x* axis indicates which scRNA-seq dataset the underlying cell-type specificity matrix was derived from. For each association, the following asterisks are overlaid to indicate the multiple test correction threshold passed: *0.05/ 12 < *P* < 0.05; **0.05/12*6 < *P* < 0.05/12; ***P* < 0.05/12*6.

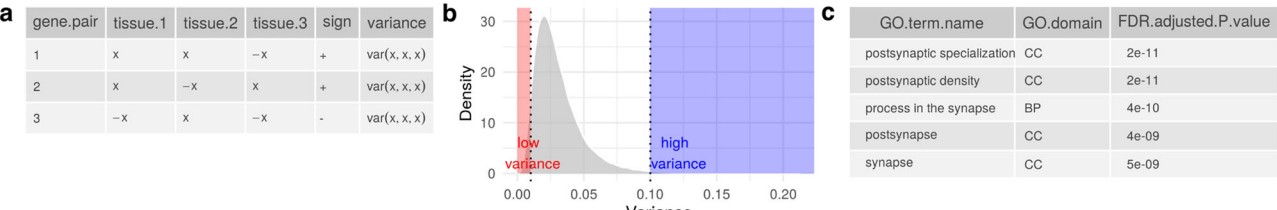

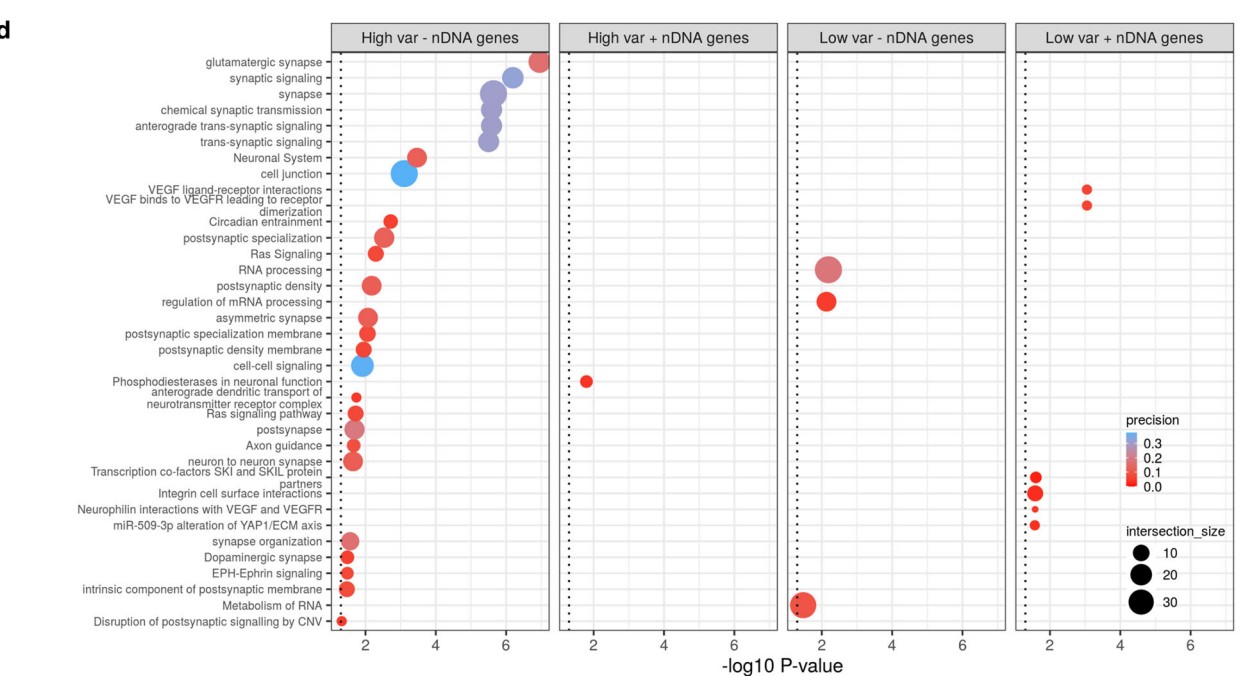

**Fig. 3 Visualisation of cross-CNS gene pair variances in GTEx data and the processes enriched in four variance-defined gene sets. a** Schematic to visualise generation of cross-CNS variances. For each mitochondrial−nuclear gene pair, a variance is taken of its per-tissue Spearman's ρ values. The pair is assigned a directionality (sign) based on the majority directionality of its ρ values. **b**. Density plot of the distribution of cross-CNS mitochondrial−nuclear gene pair variances. The left-hand dotted line enclosing the shaded red area is the cut-off for 'low variance' gene pairs, the right-hand dotted line enclosing the blue shaded area is the cut-off for 'high variance' gene pairs. **c** SynGO (synaptic gene ontology) output showing the top five enrichments for the high variance negative nuclear gene set. P values are FDR-adjusted. **d** gProfiler2-derived enrichments for four nuclear gene sets: high variance negative, high variance positive, low variance negative and low variance positive. The dotted line indicates a 5% significance cut-off. P values were corrected using the gProfiler g:SCS method, optimised for enrichment analysis P value correction.

variation. To this end, we calculated the variance of Spearman correlation coefficients of each mitochondrial−nuclear gene pair across the 12 GTEx CNS regions, and assigned correlation directionality to each pair (see example in Fig. 3a). To reduce redundancy, aggregation of mitochondrial genes was performed, taking the median cross-CNS variance of 13 mitochondrial genes as the representative value for each nuclear gene (Supplementary Fig. 3B). Using this methodology, four gene sets were defined: (1) 'high variance positive': top 5% nuclear genes with the most variable relationships with the mitochondrial genome across brain regions (N = 605); (2) 'high variance negative': top 5% nuclear genes with the most variable relationships with the mitochondrial genome across brain regions (N = 145); (3) 'low variance positive' (N = 387); (4) 'low variance negative' (N = 387). These gene sets were used as input for the gene ontology enrichment tool gProfiler2 to derive enriched pathways[25].

Overall, the distribution of variances was highly skewed towards zero, demonstrating that the vast majority of mitochondrial−nuclear pairs are stably correlated across all CNS regions (Fig. 3b). In gene pairs that showed consistency across brain regions, we observed enrichment for VEGF ligand−receptor interactions in the positive correlation set (set 3 above, P = 8.12e−04, corrected for multiple tests), whereas RNA processing (set 4 above, P = 7.72e−3, corrected

for multiple tests) was enriched in the negative correlation set (Fig. 3d). Amongst the nuclear genes with the most variable relationships to the mitochondrial genome across brain regions, we observed enrichment of phosphodiesterases in neuronal function as the only significant term for the positive (set 1 above) and synaptic terms in the negative set (set 2 above), with the most significant term being glutamatergic synapse (P = 1.42e−06, corrected for multiple tests) (Fig. 3d). To explore this enrichment further, we utilised SynGO, a specialist synapse ontology enrichment tool[26] and found significant enrichment in the high variance negative list only. This set was highly significantly enriched for postsynaptic terms (P = 3.4558e−20, FDR-corrected) with 3/5 of the most significant terms relating to this structure (Fig. 3c). Of the 28 significant terms, 13 related to 'postsynaptic' structures or processes and 5 related to 'presynaptic' (Supplementary Table 1). Overall, this analysis identified sub-cellular specificity in mitochondrial−nuclear correlations across the CNS. More specifically, variable mitochondrial−nuclear relationships highlighted genes associated with postsynaptic processes.

**Correlation magnitude, directionality and cell-type enrichment replicate in an independent dataset**. To determine whether patterns of mitochondrial−nuclear correlation observed in GTEx

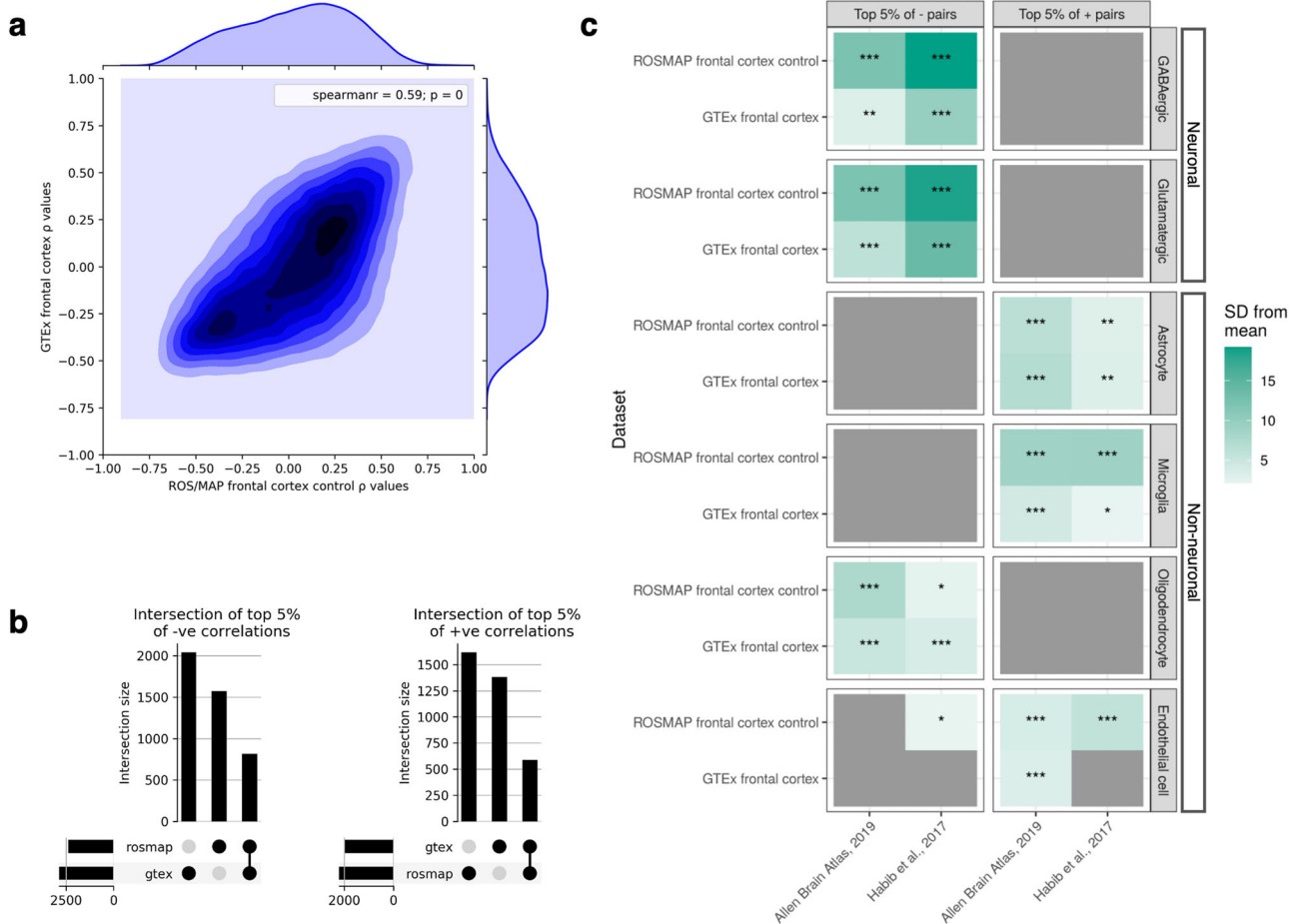

**Fig. 4 Replication of the mitochondrial−nuclear correlation values and cell-type enrichments discovered in GTEx frontal cortex in an independent frontal cortex dataset (ROSMAP control samples). a** Density-contour plot to show all mitochondrial−nuclear gene pairs commonly expressed in both datasets (177,320). ROSMAP ρ values are plotted on the x axis, and GTEx ρ values are plotted on the y axis. The Spearman correlation for the overall bi-dataset correlation and corresponding P value for the r statistic is given in the top right of the plot (Spearman's ρ = 0.59, P < 2e−16). **b** Upset plots to show numbers of unique nuclear genes found in the top 5% positive (left-hand plot) and top 5% negative correlations in the two datasets, and the overlap size of these gene sets. 817 nuclear genes were found in the top 5% of negative pairs for both datasets, and 588 nuclear genes were found in the top 5% of positive pairs for both datasets. Thus, 52% and 36% of unique nuclear genes from negative and positive mitochondrial−nuclear pairs discovered in GTEx replicate in the ROSMAP control dataset. **c** EWCE-derived cell-type enrichments for GTEx frontal cortex and ROSMAP frontal cortex. The y axis denotes the RNA-seq source. For each association, the following asterisks are overlaid to indicate the multiple test correction threshold passed: \*0.05/12 < P < 0.05; \*\*0.05/12\*6 < P < 0.05/12; \*\*\*P < 0.05/12\*6.

brain data were robust, we considered mitochondrial−nuclear gene expression correlations in neurological control samples from the ROSMAP dataset. Since ROSMAP data are derived from dorsolateral prefrontal cortex tissue, we compared the findings to those generated from the GTEx frontal cortex tissue only.

Overall, Spearman's ρ values for all mitochondrial−nuclear gene pairs showed high correlations between GTEx and ROSMAP control data (Spearman's ρ = 0.59, P < 2e−16, for 13,640 nDNA genes that were expressed in both datasets), highlighting the consistency of mitochondrial−nuclear relationships in the brain (Fig. 4b). Visual inspection of correlation distributions across the two datasets revealed greater similarity at high Spearman's ρ magnitudes, likely due to the greater accuracy associated with those correlation magnitudes (Fig. 4b). Next, we analysed the replicability of the top 5% (ranked by Spearman correlation magnitude) positively and negatively correlated gene pairs. We found that 817 nuclear genes were in the top 5% of negative pairs for both datasets, and 588 nuclear genes were found in the top 5% of positive pairs for both datasets (Fig. 4b). As such, 36% (top 5% positive) and 52% (top 5% negative) of the GTEx-derived gene sets are composed of the same genes when derived from ROSMAP data.

Given these findings, we extended replication analyses to look for evidence that the cell-type-specific enrichments identified in GTEx frontal cortex are robust across datasets. Repeating the EWCE analysis (see 'Methods') using the top 5% positive and negative gene lists generated from the ROSMAP control data (Fig. 4c), we find near-identical patterns of cell-type enrichment to GTEx data. We observed significant enrichment of genes with high neuronal specificity in negatively correlated mitochondrial −nuclear pairs (P < 0.05, Bonferroni-corrected for regions and gene sets) (Fig. 4c). There was also significant enrichment of genes with high specificity to astrocytes (P < 0.05, Bonferroni-corrected for regions) and microglia (P < 0.05, Bonferroni-corrected for regions and gene sets) amongst positively correlated mitochondrial−nuclear pairs. Enrichment of oligodendrocyte marker genes in negative pairs was also replicated in the ROSMAP frontal cortex data (P < 0.05, Bonferroni-corrected for regions and gene sets) (Fig. 4c). Thus, we see robust replication of EWCE cell-type enrichments in the ROSMAP data, where neuronal enrichment in the negative mitochondrial−nuclear space, and glial enrichment in the positive space are highly reproducible.

**Nuclear genes strongly implicated in ND have non-random relationships with the mitochondrial genome.** Given the robust nature of mitochondrial−nuclear relationships and their association with specific cell types in CNS tissue, we aimed to investigate whether genomic cross-talk is relevant to the aetiology of NDs. To this end, we tested whether mitochondrial−nuclear correlation distributions for genes implicated in NDs were significantly different to distributions generated using random sets of matched genes (Fig. 5).

We first tested four gene sets: two sets derived from AD[27] and PD[28] GWASs respectively (implicating genes through analyses of common variants), a gene set from the Genomics England PanelApp containing genes implicated in rare Mendelian forms of adult onset neurodegenerative disease, and a second PanelApp list, intracerebral calcification disorders[29], as a negative control. These were largely independent sets, with very little overlap in the genes included (for visualisation of gene set overlaps, see upset plot in Supplementary Fig. 4A). We found that genes associated with AD through GWAS analyses had mitochondrial−nuclear correlations which were nominally different (did not pass multiple test correction) from random gene sets in cortex ($P = 0.0206$) and substantia nigra tissues ($P = 0.0273$) (Fig. 5a). Similarly, a nominally significant distribution shift was observed in hypothalamus tissue using the gene set implicated in sporadic PD ($P = 0.0163$). In contrast to this, we found that genes associated with adult onset ND had highly significant differences in mitochondrial−nuclear correlations in the majority of CNS regions ($P < 0.05$, Bonferroni-corrected for regions and gene sets).

To test whether these findings were specific to a subset of NDs, we also investigated mitochondrial−nuclear correlations among genes implicated in intracerebral calcification disorders (ICDs). This disease gene set was used as a negative control since, unlike AD and PD, ICD-induced neurodegeneration is caused by calcium deposition in the brain's vasculature or parenchyma. We found no significant difference between this gene set and empirical distributions in any CNS tissues. In light of the cell-type enrichment data, in which neurons were enriched in negative pairs, this may reflect the presence in these lists of nuclear-encoded genes involved in neuronal processes.

The PanelApp adult onset ND gene set is an umbrella set, incorporating genes in the smaller and more specific 'early onset dementia' and 'PD and complex PD' PanelApp gene sets (for visualisation of overlaps, see upset plot in Supplementary Fig. 4b). As such, we aimed to look at whether these more specific disease-related subsets also had significant relationships to the mitochondrial genome. We subsequently expanded the analysis to include these gene sets, and set more stringent significance cut-offs to consider the increased number of tests. We found that genes implicated in Mendelian forms of PD (PanelApp 'PD and complex PD') showed significant differences in mitochondrial −nuclear correlations in 7/12 brain regions ($P < 0.05$, Bonferroni-corrected for regions and gene sets), including the basal ganglia ($P < 0.05$ for putamen, caudate and nucleus accumbens basal ganglia, Bonferroni-corrected for regions and gene sets) which are among the most disease-relevant tissues (Supplementary Fig. 4c). Similarly, genes associated with early onset dementia were found to have significant differences in mitochondrial−nuclear correlations in the majority of regions ($P < 0.05$, Bonferroni-corrected for regions and gene sets).

We note that in all cases, the ND-associated nuclear genes had more negative correlations with mitochondrial gene expression than would be expected by chance. Interestingly, we observed that among the ND-implicated genes with the strongest mitochondrial −nuclear correlations was *APP* (in the top 1%, ranked 54/5898 of the negative mitochondrial-nuclear pairs), which encodes the precursor protein whose proteolysis generates amyloid beta (Aβ),

the primary component of amyloid plaques. As well as this, we note that highly significant mitochondrial−nuclear relationships were observed for some genes confidently associated with complex PD[28], such as *PSAP* (Supplementary Fig. 5b). Interestingly, in *PSAP* knockout iPSC lines ROS production was seen to increase compared to controls[30]. As such, our identification of high mitochondrial-*PSAP* association lends support to this gene being important in core mitochondrial processes such as ROS-production.

Taken together, we conclude that expression levels of genes causally implicated in a subset of NDs show stronger relationships with mitochondrial gene expression than expected by chance. This analysis can be performed with a user-specified gene list using our accompanying tool available at https://ainefairbrotherbrowne.shinyapps.io/MitoNuclearCOEXPlorer/.

**Synaptic processes are enriched in mitochondrial−nuclear pairs that display correlation disparities between AD and control samples.** Finally, we analysed mitochondrial−nuclear correlations in post-mortem brain samples originating from individuals with Alzheimer's disease and from matched neurological controls. The data were covariate corrected in the same way as above, but with the addition of Scaden-derived cell-type proportions to account for disease-induced changes in cell-type density. We then calculated the difference in the correlation values between cases and controls for each mitochondrial−nuclear gene pair to produce case−control delta scores (Δρ) (Fig. 6a).

High levels of consistency between case and control mitochondrial−nuclear correlation values were observed, with 76% of pairs displaying a Δρ of <0.1 (Fig. 6b). However, we noted the presence of gene pairs displaying high delta scores, where co-expression of a pair had shifted in AD samples relative to controls (Fig. 6b). Given that we had corrected for changes in cell-type proportions, these shifts likely represent disease-associated disruptions in mitochondrial−nuclear co-expression that have the potential to drive to AD pathogenesis. To understand whether nuclear genes involved in specific biological processes were represented amongst mitochondrial−nuclear gene pairs with high delta scores, we applied Gene Set Enrichment Analysis (GSEA). First, gene pairs were split by their mitochondrial−nuclear correlation directionality, with the intuition that positive and negative correlations are representative of distinct transcriptional control mechanisms. Notably, 1.1% of significant shifts were observed among genes that switched directionality (Fig. 6a), and as such these were excluded from the analysis. This yielded two gene sets (−Δρ and +Δρ scores), which were then ranked by their absolute Δρ score (Fig. 6a).

In the negative correlation set, using fGSEA we detected 55 significant enrichments. The three most significant terms were synapse ($P = 3.5e−04$, Bonferroni−Hochberg (BH) corrected), neuron to neuron synapse ($P = 4.6e−03$, BH-corrected) and cell projection organisation ($P = 4.6e−03$, BH-corrected), detected among gene pairs that display stronger relationships in case samples compared with controls. Three of the 55 enrichments (vacuolar lumen, and lysosomal lumen and lipoprotein metabolic process) were detected among gene pairs with negative mitochondrial−nuclear correlations that show weaker association in AD samples compared with controls. Within these sets, individual genes of specific interest for AD showed particularly large absolute Δρ scores. First, *MTLN* (rank 69/14,327 gene pairs with mean correlation taken across 13 mitochondrial genes, ranked in the top 0.5% of Δρ values) encodes a protein product that is known to localise to the mitochondrial inner membrane, where it influences protein complex assembly and modulates respiratory efficiency, impacting on respiration rate, Ca2+

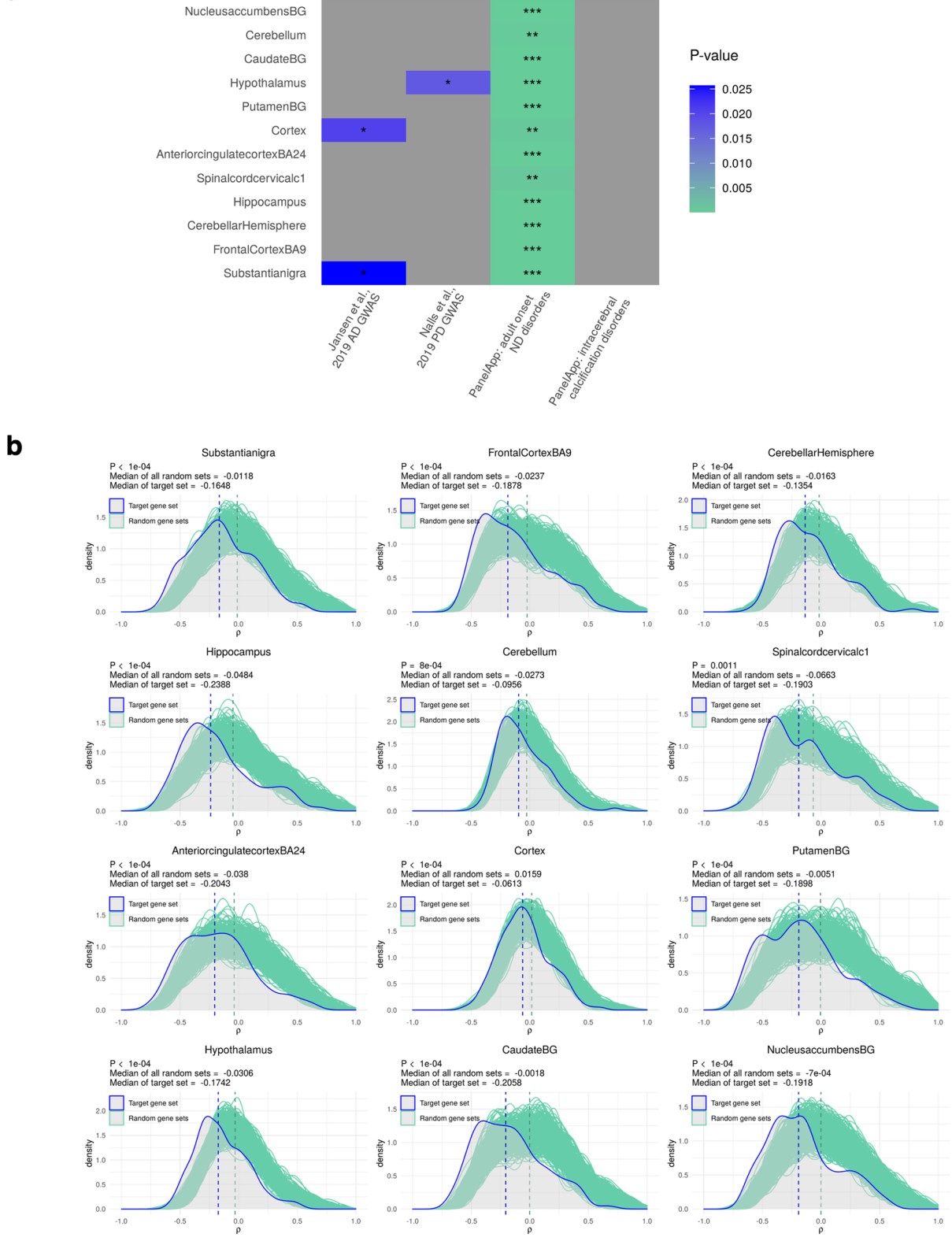

**Fig. 5 Visualisation of ND gene set associations with the mitochondrial genome. a** Heatmap to show *P* values associated with the median of four ND-related gene sets being more extreme than that of 10,000 random gene sets in 12 GTEx CNS regions. Raw *P* values (below *P* < 0.05) are represented by the colour scale, with the following asterisks overlaid to indicate which multiple test correction thresholds are passed: *0.05/12 < *P* < 0.05; **0.05/12*4 < *P* < 0.05/12; ***P* < 0.05/12*4. Grey squares indicate associations for which *P* > 0.05. **b** Visualisation of the results in part a, for the AOD target set only. The target gene set distribution is shown in blue and the distribution of 10,000 random size-matched gene sets is shown in green. Vertical dotted lines represent the medians of the target gene set (blue) and the central median of the 10,000 bootstrap sets (green). This figure was produced using the MitoNuclearCOEXPlorer tool.

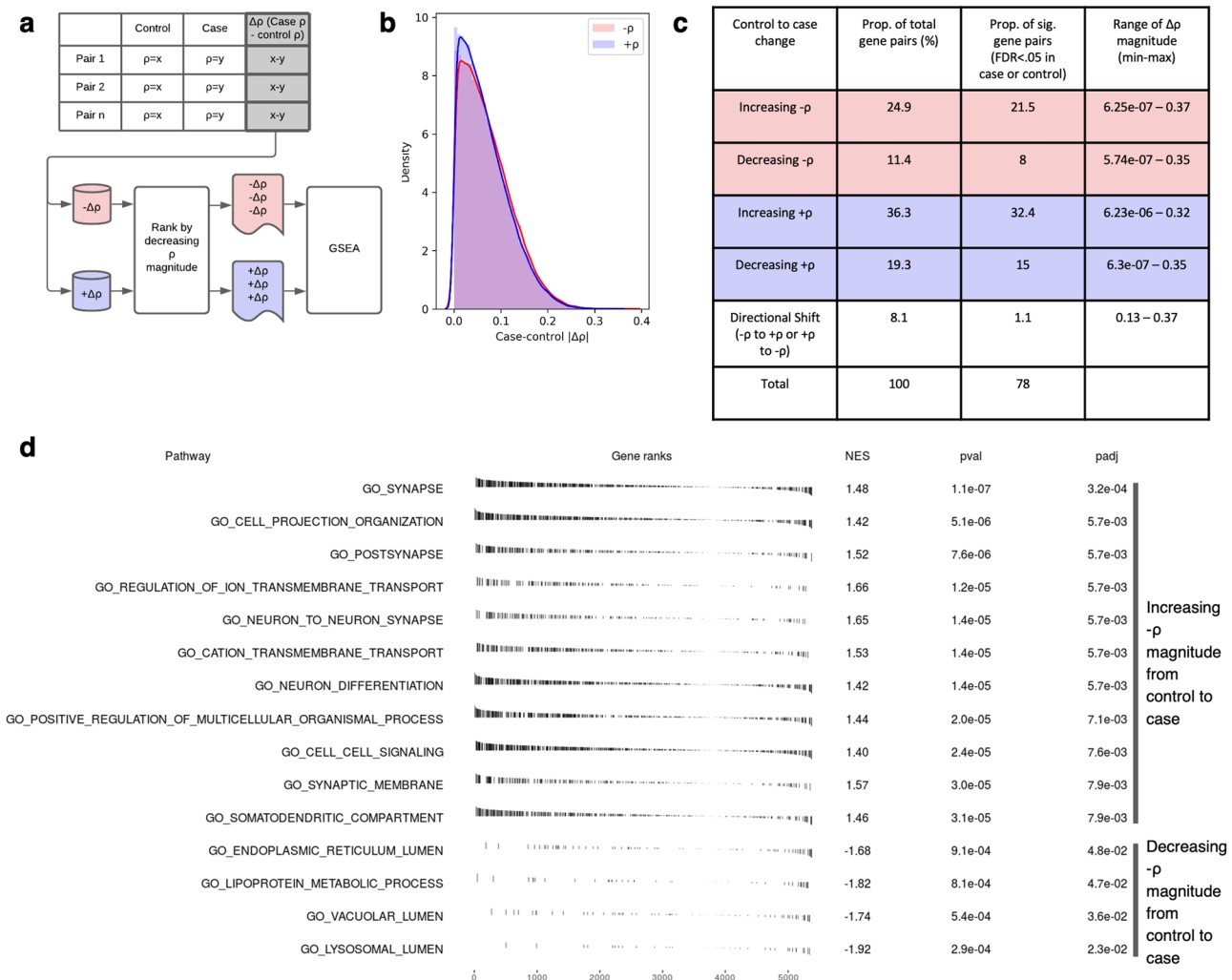

**Fig. 6 ROSMAP case−control analysis of ρ value differences (Δρ). a** Schematic to show generation of the case−control Δρ values, splitting of the data into positive and negative Δρ values and subsequent ranking strategy applied prior to GSEA analysis. **b** Density plot to show the distribution of mitochondrial−nuclear case−control Δρ in ROSMAP frontal cortex data. The red curve represents negative Δρ and the blue curve represents positive Δρ values. **c** Table to show the distinct groups of case−control Δρ values arising from the ROSMAP frontal cortex case−control data. **d** fGSEA pathway enrichments passing P < 0.05 (BH-corrected) for the negative correlation space, whereby gene pairs with −ρ have been ranked by their case−control Δρ.

retention capacity and ROS[31,32], making it of particular interest in a disease context. Second, *PSAP* (max Δρ = 0.13, mean Δρ = 549/4653 ranked in the top 12% of decreasing −Δρ values) is a leading-edge gene for the lysosomal lumen enrichment and also displays highly significant mitochondrial−nuclear relationships across brain regions (Supplementary Fig. 5). This gene is of interest in the context of AD due to its known anti-inflammatory and neuroprotective roles[33], as well as its identification as a biomarker of preclinical AD cases, enabling discrimination from control samples[34]. No enrichments reaching BH significance were detected in the positive correlation list.

## Discussion
In this work, we investigate mitochondrial−nuclear coordination in CNS tissue across the human brain. We find that CNS regional cell-type composition contributes to regional variation in co-expression, reflecting functional specialisation, specifically at synapses. Using an independent frontal cortex dataset, we show high replicability of mitochondrial−nuclear correlation distributions and cell-type-specific correlation profiles. We find that

nuclear genes causally implicated in Parkinson's and Alzheimer's disease (AD) show much stronger relationships with the mitochondrial genome than expected by chance, and that mitochondrial−nuclear relationships are highly perturbed in AD cases, particularly those involving synaptic and lysosomal genes.

A key finding of this study was the identification of cell-type as a contributor to the distinct patterns of mitochondrial−nuclear co-expression across CNS regions. Neuronal markers were enriched in negative mitochondrial−nuclear correlations, in contrast to glial (astrocytic and microglial) markers which were enriched in positive correlations. Additionally, we observed a reduction of cross-CNS variation in mitochondrial−nuclear correlations when correcting for cell-type proportions, indicating that depletion of cell-type-specific signals reduces the regional specificity of mitochondrial−nuclear relationships. Notably, correction for cell-type significantly, but not entirely, depletes cross-regional variation, indicating that although cell-type is a significant contributor, there are additional drivers of cross-regional variation in mitochondrial−nuclear relationships.

The finding that cell-type significantly contributes to regional variation in mitochondrial−nuclear association could be

explained by cell-type-specific mitochondrial specialisation. Our analysis assays a proxy for the nuclear association with ATP synthesis, and so captures a single aspect of mitochondrial function. In fact, mitochondria have many other important roles in cells, such as calcium buffering, which may vary across different cell types. As such, the division of mitochondrial−nuclear correlation directionality between cell types could be the result of divergent functionality among the mitochondria of these cell types. This is a view supported by proteomic cell-type-specific profiling of brain mitochondria. Recent work has revealed notable molecular and functional diversity of mitochondria across cell types, with astrocytic mitochondria found to perform the core cellular functions of long-chain fatty acid metabolism and calcium buffering with greater efficiency than mitochondria in neural cell types[35]. Another linked explanation for cell-type-specific correlation directionality is that it is driven by core differences in energy management strategies between cell types. In energetically demanding cell types such as neurons, anti-correlation could reflect the need for tighter OXPHOS regulation to protect against excessive ROS production, with post-transcriptional processes perhaps being used to manage local, flexible regulation of energy supply. Interestingly, oligodendrocytes are the exception among glia, displaying neuron-like enrichment in negative mitochondrial−nuclear correlations. In this context, it is worth noting that while oligodendrocyte metabolism is poorly understood, their central role in myelin sheath production is highly energy intensive, mirroring the high energy requirements of neurons[36,37].

The synapse is the site of greatest energy expenditure in the neuron[38]. To match energy supply and demand, the mitochondria in synaptic compartments display structural, biochemical and spatial plasticity[38]. To achieve this necessitates equally flexible maintenance of the mitochondrial proteome, the exact mechanisms of which are not known[38]. Our analysis reveals variable mitochondrial−nuclear relationships being highly significantly enriched for synaptic marker genes, meaning that nuclear-encoded synaptic gene expression and mitochondrial-encoded gene expression are differentially associated across the CNS. We considered the possibility that we may simply be tagging variability in regional mtDNA expression; however, residual TPM values for the 13 mtDNA genes demonstrate consistent cross-CNS expression (Supplementary Fig. 2A), suggesting that this is not a core driver of the regional specificity of mitochondrial−nuclear correlation profiles. It could be that we are indirectly observing mitochondrial plasticity by capturing neuronal subtype-specific variation in nuclear and mitochondrial expression. It is known that neuronal subtypes are energetically specialised[39], and that CNS tissues have differential neuronal subtype compositions[40,41]. Sub-cell-type-specific expression modulation as a mechanism to manage local energy requirements at synapses is supported by work finding that heterogenous energy requirements across CNS regions and cell types may necessitate bespoke mitochondrial proteomes[38]. Further to this, molecular evidence shows that several nuclear-encoded mitochondrial genes involved in processes key to mitochondrial plasticity (mitochondrial transcription, fission and trafficking) have been found to exhibit distinct patterns of expression in the neuronal subtypes[42]. Recent work using engineered MitoTag mice coupled with an isolation approach to profile tagged mitochondria from defined cell types has demonstrated profound cell-type-specific mitochondrial biology serving homoeostatic needs to preserve essential functions in cells[35]. And yet, without directional information and cell-type or sub-cell-type-specific data, it is difficult to make a firm assertion as to whether the underlying mechanism is anterograde modulation of the mitochondrial genome from the nucleus, or retrograde modulation of the nuclear genome by the mitochondria, or perhaps a feedback loop involving both.

Uniquely to the field of mitochondrial−nuclear cross-talk, we look at its genome-wide relevance with respect to a range of neurodegenerative diseases. Testing the association of ND-implicated genes with the mitochondrial genome demonstrated significant non-random correlations between mitochondrial gene expression and ND-implicated nuclear genes. While genes implicated in PD and AD through GWAS analyses showed nominally significant associations with the mitochondrial genome, it should be noted that there are likely to be inaccuracies in variant-gene assignments within these sets which weaken the analysis. Interestingly, this view is supported by high confidence enrichments of mitochondrial−nuclear association in nuclear gene sets associated with Mendelian forms of the same diseases. Mendelian AD and PD genes displayed highly significant shifts from random, all of which were towards higher negative correlation magnitudes, and highlighted particularly strong correlations among important ND genes. In fact, APP, the first gene to be causally implicated in AD, ranked in the top 3% of all pairs with negative associations.

Given these findings, we postulated that analysing changes in mitochondrial−nuclear correlations in the context of AD would provide important disease insights. To do this, we leveraged the AD case−control ROSMAP dataset. After correcting for cell-type proportion, we observed an enrichment of synaptic terms among nuclear genes which were negatively correlated with mitochondrial gene expression and which had stronger relationships in the context of AD than in control samples (i.e. high case−control correlation difference, $\Delta\rho$, gene pairs). Given the close relationship between synapses and mitochondria, with multiple lines of evidence pointing not only to synaptic function being dependent on mitochondria, but to mitochondrial regulation of synaptic plasticity[43–45], the tightening co-expression here could represent a drive to recover energetic homoeostasis at damaged synapses and increase their efficiency. In support of this, we see that the mitochondrial efficiency enhancing gene MTLN[31] is in the top 1% of increasing negative associations. In particular the MTLN-MTCYB gene pair displayed a striking $\Delta\rho$, where in control samples the pair had a non-significant correlation ($\rho = -0.008$, $P = 0.93$), but shifted to a highly significant association with a considerably higher negative magnitude in case samples ($\rho = -0.27$, $P = 3.01e-05$).

Interestingly, we also observed enrichment of lysosome-related terms (lysosomal lumen, vacuolar lumen) in negatively correlated gene pairs that weaken in case samples relative to controls (Fig. 6d). Lysosomes are essential for the removal of dysfunctional mitochondria as well as other organelles and proteins, and there is growing evidence to suggest that lysosomal dysfunction contributes to the pathogenesis of AD[46,47], as well as PD[48]. Perhaps decoupling of nuclear genes in these pathways from mitochondrial gene expression represents a reduction in the efficacy of dysfunctional mitochondria clearance, thus augmenting the pathology.

These results provide more evidence for the role of mitochondria in neurological disorders, and identify particular pathways and processes that may be more relevant to the aetiology of disease. As such, targeting these routes to dysfunction may be particularly fruitful for the treatment of specific neurological disorders.

## Methods

**GTEx data**. Raw RNA-sequencing data from 12 histologically normal CNS regions were obtained from the Genotype-tissue Expression project (GTEx, V6p)[49]. Processing was carried out as per ref. [18]. Briefly, adapter sequences, low-quality terminal bases and poly-A tails (>4) were trimmed and subsequently aligned to the 1000G GRCh37 reference genome using STAR. Strict filtering was applied to avoid misalignment of NUMT sequences, and to retain only properly paired and uniquely mapped reads. Post-mapping processing included exclusion of samples with: <10 K

reads mapping to the mitochondrial genome, <5 million total mapped reads, >30% of reads mapping to intergenic regions, >1% total mismatches or >30% reads mapping to ribosomal RNA using custom scripts as well as RNAseQC. HTseq was used to quantify transcripts, before converting raw counts to TPMs using version 19 of the Gencode gene annotation. The final per-brain-region sample (n) numbers and number of genes expressed are shown in Supplementary Table 2.

**ROSMAP data.** The ROSMAP dataset is composed of dorsolateral prefrontal cortex samples derived from autopsied individuals from the Religious Orders Study (ROS) and the Rush Memory and Aging Project (MAP)[50]. Data were obtained through application to the data access committee, permitting access to pre-mapped FPKM data (for QC and mapping details see ref. [42]). Each ROSMAP sample is associated with a cognitive diagnosis. We used samples labelled 'AD' (n = 254) and 'no cognitive impairment' (n = 201), referred to as 'case' and 'control', respectively. Samples with missing metadata and duplicates were removed, reducing the number of cases to 251. Prior to further processing, FPKMs were converted to TPMs.

**Generating mitochondrial−nuclear correlation matrices.** For both datasets, the same custom pipeline was applied to generate mitochondrial−nuclear gene expression correlation matrices from gene counts. First, TPM matrices were filtered for genes with TPM > 0 in all samples, and samples with TPM = 0 in all genes were removed. TPMs were then log10 and median normalised. Expression outliers, defined as TPM values three interquartile ranges below the lower quartile or above the upper quartile for a gene, were masked.

Covariates for data correction were selected by performing principal component analysis (PCA) on the expression matrices. Spearman correlations between the largest axes of variation (first ten principle components, capturing 98.41% of the variation for GTEx and 99.43% for ROSMAP) and known covariates were calculated (Supplementary Fig. 6). For ROSMAP, the following covariates were selected: PMI, RIN, library batch, race, sex, study, age at death, age at last visit. For GTEx, the following covariates were selected: RIN, four batch variables (type of nucleic acid isolation batch, nucleic acid isolation batch ID, genotype or expression batch ID, date of genotype or expression batch), centre, age, gender and cause of death.

Following this, multiple linear regression was applied to regress out covariates. TPM values were included as predictor variables and covariates as response variables in a linear model. Predicted TPMs were calculated following model fitting, and residuals were calculated by subtracting predicted from observed, yielding residual TPMs. To generate mitochondrial−nuclear correlation matrices, Spearman correlation coefficients were calculated between protein-coding mitochondrial genes (13) and nuclear genes (for GTEX: 15,001 genes expressed in all CNS tissues; for ROSMAP all nuclear genes expressed).

**Analysing mitochondrial−nuclear correlation variance across CNS regions.** To understand the extent to which the relationships between mitochondrial−nuclear gene pairs vary across the CNS, we leveraged 12 GTEx CNS regions, calculating a cross-CNS variance of correlation coefficients for every mitochondrial−nuclear gene pair (see tabular method schematic in Fig. 3b). We then calculated the variance of these 12 coefficients as a measure of variation in the relationship between the expression of these two genes across CNS regions. We repeated this for all mitochondrial−nuclear gene pairs. Nuclear genes expressed in all 12 CNS regions were used, equating to 15,001 nuclear genes and 195,013 mitochondrial−nuclear pairs. To reduce redundancy of the dataset, aggregation of mitochondrial genes was performed, the intuition being that the correlation of a nuclear gene with the 13 mitochondrial genes was found to be largely consistent (Supplementary Fig. 3B). The median cross-CNS variance of 13 mitochondrial genes was taken as the representative value for each nuclear gene.

To determine processes enriched in gene pairs in different variance brackets, four gene sets were defined. The 'high variance set' (highest 5% of variances, n = 750), and the 'low variance set' (lowest 5% of variances, n = 750). These two groups were then further split into positive and negative sub-groups, dependent on the majority correlation directionality. This yielded the following gene sets: high variance positive, n = 605; high variance negative, n = 145; low variance positive, n = 363; low variance negative, n = 387.

To determine the processes and pathways enriched in these gene sets, the R package gProfiler2 was used. Enrichments were tested against a custom background of genes expressed in all GTEx CNS regions (n = 15,001). The queries were ordered by correlation magnitude, and for multiple test correction, the 'g:SCS' method was applied. Enriched terms were visualised in barplots. To obtain a more granular ontology analysis of the synaptic enrichment observed in the high variance negative group, this list was used as input to the online tool SynGO[26]. The same background list was used for SynGO as for gProfiler2.

**EWCE analysis.** Expression Weighted Cell-type Enrichment (EWCE) was used to determine whether nuclear gene sets had higher expression within particular CNS cell types than would be expected by chance[20]. EWCE leverages single-nuclear RNA-seq (snRNA-seq) data in the form of specificity matrices. Specificity matrices give, for each gene and each cell-type, the expression specificity a gene has in a cell-type compared with all other cell types. Using this information, EWCE statistically evaluates whether cell-type-specific markers have higher expression in a target list

than would be expected by chance (i.e. than the random distributions drawn from the background).

Inputs to EWCE were target gene lists, a background gene set and a specificity matrix. Aggregation over mitochondrial genes was then performed (as above) to obtain a single consensus ranking for each nuclear gene. The target gene lists used were generated by ranking mitochondrial−nuclear correlation values for each GTEx CNS region with the largest positive and negative values ranked separately. For each CNS region, the top 5% of positively correlated nuclear genes and top 5% of negatively correlated nuclear genes were then taken as region-specific target gene sets. The numbers of genes per region are given in Supplementary Table 2. The background gene set was genes expressed in all GTEx CNS regions (n = 15,001). Specificity matrices were generated as per Skene et al.[20] by estimating the specificity of each gene to each cell-type. The specificity score represents the proportion of the total expression of a gene found in one cell-type compared to all cell types. Data used to generate specificity matrices for this work were derived from two brain snRNA-seq experiments. (1) The Allen Brain Atlas: a dataset comprising 15,928 nuclei from the middle temporal gyrus of 8 human tissue donors ranging in age from 24 to 66 years[51]. (2) Habib[22]: a dataset comprising 19,550 nuclei from the hippocampus (4 samples) and prefrontal cortex (3 samples) from five donors.

The EWCE analysis was run with 10,000 bootstrap lists. Transcript length and GC-content biases were controlled for by selecting bootstrap lists with equivalent properties to the target list. P values were corrected for multiple testing using the Benjamini−Hochberg method over all cell types and gene lists tested. We performed the analysis with major cell-type classes ('GABAergic', 'glutamatergic', 'astrocyte', 'microglia', 'oligodendrocyte', 'endothelial cell').

**Cell-type correction analysis.** To further evaluate the contribution of cell-type to heterogeneity of mitochondrial−nuclear correlation distributions in the CNS, we used published cell-type proportion estimates[24] to understand whether correcting the GTEx expression data for the effect of cell-type proportions would result in more homogenous mitochondrial−nuclear correlation profiles across the CNS. To do this, we compared the effect on mitochondrial−nuclear correlation distributions of two correction strategies, one which does not correct for cell-type, and one which does: (i) standard correction (covariates: RIN, four batch variables (type of nucleic acid isolation batch, nucleic acid isolation batch ID, genotype or expression batch ID, date of genotype or expression batch), centre, age, gender and cause of death), (ii) standard + cell-type correction (covariates: as in (i) as well as the proportions of the following cell types: astrocyte of the cerebral cortex, Bergmann glial cell, brain pericyte, endothelial cell, neuron, oligodendrocyte and oligodendrocyte precursor cell). The sample set used for the calculation of mitochondrial−nuclear correlations differed to that used for the other analyses presented in the paper due to availability of cell-type proportion data (sample numbers per tissue provided in Supplementary Table 3). Upon filtration for these samples, the pipeline as described previously, was run, producing mitochondrial−nuclear correlation matrices for both correction strategies.

This analysis relied on the assumption that the cell-type proportions were representative of the cellular composition of GTEx CNS regions. However, there are several factors that affect the accuracy of the proportion estimates. Firstly, they were derived using murine brain scRNA-seq data, and so there is a species mismatch here. Secondly, technical factors such as the dissection protocol (excision order and resulting RNA degradation), the size of each target region and the accuracy of tissue excision. Assignment of cell-type proportions to the GTEx spinal cord tissue, which was not dissected from the mice, was present in the cell type proportion data, and as such, these proportions cannot be considered to accurately represent the cell-type composition of the spinal cord. Considering these factors, we looked to select a subset of regions that were most accurately represented by the cell-type proportion estimates. We calculated 20 axes of variation from GTEx CNS RNAseq data using PEERv1.0[52] on all samples, obtaining 20 PEER factors for each region. We correlated these with the cell-type proportion estimates to understand whether the cell-type proportions represented the cell-type composition of each GTEx CNS region. The underlying logic being that larger axes of variation would correlate strongly with the cell-type proportion estimates if those estimates accurately represented that tissue.

We identified five CNS regions (anterior cingulate cortex, cortex, frontal cortex, hippocampus, caudate basal ganglia) for which PEER factors explaining most variance in the data were correlated with cell-type proportions. In these regions there were fewer spurious correlations with cell types across the 20 PEER factors, and PEER factors explaining most variance (PEER 1 and 2) were highly correlated (ρ ≥ 0.6) with cell types, indicating better representation of these regions by the cell-type proportion data. To obtain a measure for cross-CNS variation of mitochondrial−nuclear relationships, we calculated variances for each mitochondrial−nuclear gene pair across these five regions. We did this for correlation values produced by both correction strategies. Finally, to understand whether the cell-type correction changed cross-CNS variation in mitochondrial−nuclear correlation distributions, we performed a two-sample Wilcoxon signed rank tests with the null hypothesis that the true location shift from standard to standard and cell-type correction-derived distributions was < 0 (aka a negative shift in variance, closer to a median of zero in the latter). We also performed one-sample Wilcoxon tests for each correction strategy to test the null hypothesis that the median of the distribution of variances was equal to zero.

**Testing disease implicated gene lists against a random background**. The aim of this analysis was to determine whether specific disease-relevant gene sets had more extreme distributions of mitochondrial−nuclear gene expression correlations than a random, equally sized, set of genes. To this end, we first analysed four gene sets (sets 1, 2, 5 and 6 below), and then expanded our analysis to incorporate seven gene sets to validate our findings and further test our hypothesis with smaller, more specialised sub-lists of disease genes. The total catalogue of the seven sets used is as follows: (1) A set of 35 AD-associated genes of interest were derived from a recent AD GWAS[27]. This study analysed SNPs in clinically diagnosed 71.88K cases and 383.378K controls, identifying >20 AD-associated loci. (2) A set of 62 PD-associated genes of interest were selected on the basis of eQTL data from a recent PD GWAS[28]. This study analysed 7.8M SNPs in 37.7K cases, 18.6K UK Biobank proxy-cases, and 1.4M controls, identifying 90 signals at genome-wide significance. The Genomics England PanelApp tool gives sets of clinically curated genes associated with disease through rare variants[29]. The following panels were downloaded from this resource: (3) Early onset dementia (28 genes). (4) PD and complex PD list (43 genes). This panel contains genes associated with early onset and familial Parkinson's disease as well as complex Parkinsonism. (5) Adult onset ND disorders (110 genes). This panel is a super-set, including the early onset dementia and PD PanelApp panels as well as genes from other ND-related panels wherein mutations are known to cause ND. (6) Intracerebral calcification disorders (21 genes) used as a negative control because the pathogenesis of these disorders is distinct from AD and PD. (7) A set of genes curated by OMIM[53], including genes associated with PD phenotypes (Parkinson's disease—PS168600) (24 genes). Again, we found a highly significant shift compared to random in the OMIM set (Supplementary Fig. 4C).

For each GTEx CNS region, $r$, and each gene set, $l$, the median mitochondrial−nuclear correlation value of $l$ for $r$ was calculated. The distribution of mitochondrial−nuclear pairs was inclusive of all mitochondrial correlations for each nuclear gene, making the size of the distribution (length $l$) × 13. To generate empirical distributions, a random sample of nuclear genes of matching biotype and length, $l$, was selected from the set of genes expressed in all GTEx CNS regions (15,001) and all correlations with mitochondrial genes were included.

A two-tailed test was carried out to determine whether $l$ had a more extreme median mitochondrial−nuclear correlation value than could be expected by chance. To this end, the median of $l$ was compared to the medians of 10,000 randomly selected gene sets. $P$ values were calculated as follows, where $k$ is the number of randomly selected sets, and $n$ is the number of correlations more extreme than the median of $l$:

$$P = (k \pm n)/k$$

A series of significance thresholds of increasing stringency were calculated to reflect the number of tests that were performed, taking into consideration the number of tissues and the number of gene sets analysed. The significance of results was assessed against the following Bonferroni multiple-test corrected $P$ value thresholds: $0.05/12 < P < 0.05$; $0.05/12*$(number of gene sets) $< P < 0.05/12$; $P < 0.05/12*$(number of gene sets).

Alongside this publication, we release a tool to enable performance of this analysis with a user-specified gene list, along with single gene querying of the correlation data. This tool can be found at https://ainefairbrotherbrowne. shinyapps.io/MitoNuclearCOEXPlorer/ and the accompanying source code can be found at https://github.com/ainefairbrother/MitoNuclearCOEXPlorer[54].

**Case−control analysis of ROSMAP data**. To identify mitochondrial−nuclear gene pairs that are modulated in disease states, we used the ROSMAP case−control AD dataset. Due to cell-type proportion changes related to disease pathogenesis in AD brain tissue, we corrected for cell-type proportion in addition to the previously listed covariates using deconvolution-derived cell-type proportions[55–57]. The cell-type proportion distributions for the case and control ROSMAP data can be seen in Supplementary Fig. 7. To quantify changes in mitochondrial−nuclear co-expression, aggregation over mitochondrial genes was carried out for the case and control data separately by taking the median Spearman's ρ value for each nuclear gene. The difference between these values was then calculated (control ρ − case ρ) for each gene pair, giving case−control delta values, Δρ.

To identify pathways enriched in high Δρ values (i.e. pairs with large case−control disparities), we applied the GSEA method using the fGSEA R package[58]. The inputs into fGSEA were gene lists ranked by Δρ and split by directionality. With a separate positive and negative correlation list, the sign of the Δρ in each case relates to whether a gene pair's correlation magnitude has increased or decreased in case in comparison to control. As such, any enrichments are interpretable as being related to case−control shifts.

The fGSEA parameters used were as follows: GO as the annotation source, minimum and maximum size of terms 15 and 2000 respectively. fGSEA was run with the fgseaMultilevel function and output was visualised using the plotGseaTable function.

**Statistics and reproducibility**. Statistics were performed in R Studio (R version 3.6.3) and Jupyter Notebooks (Python 3.7). Nuclear−mitochondrial correlation matrices were derived from GTEx count matrices for 12 CNS tissues (range of $N = 43–89$), which were TPM, log10 and median normalised. The normalised TPM values were then corrected for covariates (see 'Methods') using linear regression. The residual values were then used to generate Spearman correlation matrices. $P$ values for the Spearman correlations were FDR-corrected prior to downstream analyses. The ROSMAP frontal cortex control dataset ($N = 201$) was used for replication to ensure robust and reproducible analyses.

**Reporting summary**. Further information on research design is available in the Nature Research Reporting Summary linked to this article.

## Data availability
The datasets generated and/or analysed during the current study are available through the GTEx portal (https://gtexportal.org/home/datasets, GTEx V6p, dbGaP accession phs000424.v6.p1) and the Synapse portal for ROSMAP data (https://adknowledgeportal. synapse.org/Explore/Studies/DetailsPage?Study=syn3219045). Processed GTEx data (per-tissue nuclear-mitochondrial correlation matrices) are available for download via our MitoNuclearCOEXPlorer web tool (https://ainefairbrotherbrowne.shinyapps.io/ MitoNuclearCOEXPlorer/). All data behind the main figures are available on figshare (https://figshare.com/projects/Figure_datatables_for_Mitochondrial-nuclear_cross-talk_in_the_human_brain_is_modulated_by_cell-type_and_perturbed_under_neuro degenerative_disease_status/122770).

## Code availability
All code is written in R (version 3.6.3) and Python (version 3.7). The correlation matrix-generating pipeline, written in both Python and R, is available on GitHub (https:// github.com/ainefairbrother/MitoNuclear_coexpression_pipeline)[59]. As is all code underlying the MitoNuclearCOEXPlorer tool (https://github.com/ainefairbrother/ MitoNuclearCOEXPlorer)[54].

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

## Acknowledgements

A.F.-B. was supported through the award of a Biotechnology and Biological Sciences Research Council (BBSRC UK) London Interdisciplinary Doctoral Fellowship. A.T.A. was supported by the Generation Trust. R.H.R. and Z.C. were supported through the award of Leonard Wolfson Doctoral Training Fellowships in Neurodegeneration. M.R. was supported through the award of a Tenure Track Clinician Scientist Fellowship (MR/N008324/1). This award also supported D.Z. and S.G.-R. AH holds a Medical Research Council (MRC) eMedLab Medical Bioinformatics Career Development Fellowship, funded from award MR/L016311/1. For GTEx data: The Genotype-Tissue Expression (GTEx) Project was supported by the Common Fund of the Office of the Director of the National Institutes of Health (commonfund.nih.gov/GTEx). Additional funds were provided by the NCI, NHGRI, NHLBI, NIDA, NIMH, and NINDS. Donors were enrolled at Biospecimen Source Sites funded by NCI\Leidos Biomedical Research, Inc. subcontracts to the National Disease Research Interchange (10XS170), Roswell Park Cancer Institute (10XS171), and Science Care, Inc. (X10S172). The Laboratory, Data Analysis, and Coordinating Center (LDACC) was funded through a contract (HHSN268201000029C) to the The Broad Institute, Inc. Biorepository operations were funded through a Leidos Biomedical Research, Inc. subcontract to Van Andel Research Institute (10ST1035). Additional data repository and project management were provided by Leidos Biomedical Research, Inc.(HHSN261200800001E). The Brain Bank was supported by University of Miami grant DA006227. Statistical Methods development grants were made to the University of Geneva (MH090941 and MH101814), the University of Chicago (MH090951, MH090937, MH101825, & MH101820), the University of North Carolina—Chapel Hill (MH090936), North Carolina State University (MH101819), Harvard University (MH090948), Stanford University (MH101782), Washington University (MH101810), and to the University of Pennsylvania (MH101822).

## Author contributions

A.F.-B., M.R and A.H. conceived and designed the study. A.F.-B. analysed the data, generated figures, developed the web resource and together with M.R. and A.H. wrote the first draft of the manuscript. A.T.A. processed the GTEx data from raw to read counts. R.H.R. provided code and specificity matrices for the cell-type-specific analysis. S.G.-R. assisted with the technicalities of the web app. D.Z., Z.C., R.H.R., M.R. and A.H. helped guide and troubleshoot analyses. A.F.-B., A.H. and M.R. contributed to the critical analysis of the manuscript.

## Competing interests

The authors declare no competing interests.

## Ethics approval and consent to participate

This manuscript used anonymised human transcriptomic data provided by the GTEx[40] and ROSMAP[42] projects. Both the ROSMAP and GTEx studies obtained informed consent from participants.

**Additional information**

