## [Transparent Peer Review File · Communications Biology]

Reviewers' comments:

Reviewer #1 (Remarks to the Author):

This is an omics-genetics study focused on questions of whether mitochondrial function could occupy an influential place in the AD/PD heirarchy. It generally reports data interpreted to support this view. There are some attractive efforts, for example looking at different brain regions and also looking at the enrichment of particular cell types (neurons versus the different glia components). The neuron vs glia analysis is particularly interesting. There is and impression this manuscript may generate more questions than it addresses, but it is certainly a good start in attempting to get at a worthwhile question.

Reviewer #2 (Remarks to the Author):

This paper discusses the interplay between nuclear and mitochondrially encoded genes in the brain, and how this relationship is disrupted in cell types by certain neurodegenerative diseases. The authors claim that regional differences in nuclear-mitochondrial gene expression correlation is driven by cell type distributions in these regions and show that spatially close regions have similar correlation distributions. They also show that genes associated with postsynaptic processes have the most highly variable correlations across brain regions. Finally they find that genes associated with neurodegenerative diseases are significantly more negatively correlated with mitochondrial genes than expected, as well as showing in an AD dataset that certain genes have their association is mitochondria disrupted in the disease case when compared to controls.

Overall I find this paper to be an interesting evaluation of mito-nuclear variation across the brain, of particular interest is the analysis comparing control and AD gene expression relationships. However I could not find the link between the enrichment of highly correlated gene pairs for marker genes and the claim that it is regional cell type composition that drives patterns of mito-nuclear co-expression as these enrichments were found across all regions. I found the link between the gene lists derived from PD and AD GWASs and the mitochondrial transcriptome to not be very compelling, and the remaining gene lists which show significance have significant overlap, weakening the conclusion of this section. Adding another independently derived gene list to this section would increase support for the conclusion found.

Major Comments:

1. The authors show that the most highly correlated genes are enriched for high specificity cell type marker genes, and these enrichments are a common feature to a majority of regions surveyed. The results as stated seem only to indicate that cell type is an indicator of mitochondrial expression levels, and it is not obvious to me why these results imply the shift in correlation distribution is caused by a change in cell type composition of the regions. An explanation of this conclusion would clarify the interpretation of the results.
2. The discussion surrounding highly variable nuclear-mitochondrial correlations (starting line 515) focuses on the set of variable genes being anti-correlated with mitochondrial genes, suggesting a mechanism to reduce ROS, while not referring to the variability of these relationships across brain regions implied by the increased variability of the correlations. Significant expansion of the discussion to include the implications of the variability is necessary to frame this result.
3. The heatmaps shown throughout the paper (fig 2, fig 4c, fig 5a) omit colours for non-significant results, but in testing for significance there is no mention of corrections for testing multiple gene lists or brain regions. Some results shown in these heatmaps are left as significant without consideration for the number of tests performed leaving the impression of significant results that could have arisen by chance – though the key findings mentioned in the text are found at a significance level which

accounts for this possibility, except for results on page 16. The authors should make clear which results are robust to multiple testing corrections, and whether this alters the conclusions. This is particularly important in the discussion of neurological disease gene sets, as two of the 5 relevant gene sets fail to produce any results robust to these corrections.

Minor Comments:

4. A supplementary figure showing the agreement between the nuclear-mitochondrial correlation between the 13 mitochondrial genes would provide support to the aggregation of mitochondrial genes when finding the cross-CNS variance of genes and difference in correlation of genes between AD and control.

5. Though mentioned in the methods it is unclear in the main text that the median nuclear-mitochondrial variance was used to curate high and low variance gene lists (pages 10-12)

6. When significant is indicated by asterisks (fig 2, fig 4, fig 5) the figure captions have significance marked the wrong way round (eg. * $0.05 < p < 0.05/12$, ** $0.05/12 < p < 0.05/12*6$, *** $p < 0.05/12*6$  * $0.05/12 < p < 0.05$, ** $0.05/12*6 < p < 0.05/12$, *** $p < 0.05/12*6$)

7. Line 364 references a supplementary figure 5b which is not present in the review copy.

We thank the reviewers for the thoughtful and positive comments on our manuscript entitled, "Mitochondrial-nuclear cross-talk in the human brain is modulated by cell type and perturbed under neurodegenerative disease status", and we believe that they have helped us to improve our work. We have now performed additional analyses and edits, and below you will find our detailed responses to the reviewer's comments. Changes to the main text as a consequence of these comments are tracked and all of our major conclusions remain unchanged.

Yours Sincerely,

Alan Hodgkinson

Reviewer report:

Reviewer #1

We thank you for your positive comments on the manuscript and note that no specific changes have been requested.

Reviewer #2

I found the link between the gene lists derived from PD and AD GWASs and the mitochondrial transcriptome to not be very compelling, and the remaining gene lists which show significance have significant overlap, weakening the conclusion of this section. Adding another independently derived gene list to this section would increase support for the conclusion found.

Reply: We agree that presenting multiple overlapping disease gene sets could be misleading and may appear to weaken the results. However, this cannot be solved through the provision of an independently derived gene list because it is well-recognised that many of the genes that give rise to familial forms of neurological disease are capable of causing multiple, overlapping phenotypes. Furthermore, PanelApp was chosen as a source of disease gene lists because it provides highly comprehensive and expert-curated gene lists, making it technically hard to find a truly independent source of information. Nonetheless, in order to include an additional gene set, which is at least partially independent of PanelApp, we identified the OMIM Parkinson disease list (PS168600), which contains 24 genes associated with PD phenotypes (of which 8 genes were not present in another gene set).

To streamline and simplify our analysis, we have now split it into two analytical phases: (1) the primary (main text) analysis using four non-overlapping gene sets, and (2) the secondary (supplementary) analysis using 3 additional gene sets that have varying degrees of overlap.

The primary analysis includes: i) PanelApp adult onset neurodegeneration, the largest most comprehensive PanelApp neurodegeneration set, ii) the AD GWAS set, iii) the PD GWAS set, and iv) the negative control set, consisting of PanelApp intracerebral calcification disorders.

We provide an upset plot visualising the gene overlaps for these four sets (supplementary figure 7A). We have also regenerated the main figure accordingly (figure 5). Figure 5A now only contains these four lists, and figure 5B now only contains the significant shifts attributable to CNS regions for the PanelApp adult onset neurodegeneration set to visualise the shifts in each CNS region. We have described this in the results section of the manuscript starting on page 12 as follows:

"We first tested four gene sets: two sets derived from AD and PD GWASs implicating genes through analyses of common variants, a gene set from the Genomics England PanelApp containing genes implicated in rare Mendelian forms of adult onset neurodegenerative disease, and a second PanelApp list as a negative control (intracerebral calcification disorders). These were largely independent sets, with very little overlap (for visualisation of gene set overlaps, see upset plot in supplementary figure 7A)".

The secondary (supplementary) analysis is a deep-dive, looking at disease-specific sets that overlap to various extents with the independent set included in the primary analysis (supplementary figure 7B is an upset plot visualising the gene set overlaps). We also included our OMIM gene set results here, deeming the set not independent enough to include in the primary analysis. We noted this in the methods section of the manuscript starting on page 30 as follows:

"The total catalogue of the seven sets used is as follows ... (7) A set of genes curated by OMIM⁶³, including genes associated with PD phenotypes (Parkinson's disease - PS168600) (24 genes). Again, we found a highly significant shift compared to random in the OMIM set (supplementary figure 5C)".

We also made the distinction between primary/secondary analysis clear in the results section of the manuscript starting on page 13 as follows:

"The PanelApp adult onset ND gene set is an umbrella set, incorporating genes in the smaller and more specific early onset dementia, and PD and complex PD PanelApp gene sets (for visualisation of overlaps, see upset plot in supplementary figure 7B). As such, we aimed to look at whether these more specific disease-related subsets also had significant relationships to the mitochondrial genome. We set more stringent significance cut-offs to consider the increased number of tests".

Significance thresholds for the primary and secondary analyses were adjusted to account for the number of tests carried out in each. As before, our results suggest strong and significant relationships between nuclear encoded genes causally implicated in neurological disorders and the expression of genes in the mitochondrial transcriptome.

Major Comments:

1. The authors show that the most highly correlated genes are enriched for high specificity cell type marker genes, and these enrichments are a common feature to a majority of regions surveyed. The results as stated seem only to indicate that cell type is an indicator of mitochondrial expression levels, and it is not obvious to me why these results imply the shift

in correlation distribution is caused by a change in cell type composition of the regions. An explanation of this conclusion would clarify the interpretation of the results.

Reply: We thank the reviewer for raising the interesting and important question of how to interpret the data demonstrating that the most highly correlated genes are enriched for genes with a high cell type specificity. While we agree that regional differences in the correlation distribution could be caused by a range of factors, we find evidence to support that cell type composition is an important element.

First, we note that there are consistent patterns of cell type enrichment amongst regions that are developmentally-related and structurally similar, such as cortical brain regions or regions of the basal ganglia. We have clarified this point in the results section of the manuscript, starting on page 7 as follows:

"Reassuringly, we note that related regions display similar cell-type enrichment profiles, indicative of biological functionality being reflected in these enrichments. For example, GTEx-defined⁴¹ technical sample replicates (the cortex and frontal cortex, and cerebellum and cerebellar hemisphere) as well as regions closely biologically associated such as the basal ganglia (putamen, nucleus accumbens and caudate), demonstrate consistent patterns of cell-type enrichment".

Second, and most importantly, we have now performed new analyses that indicate when gene expression data is corrected for cell type proportions, the regional differences in the correlation distributions are significantly reduced. We have now included these analyses in the results section of the manuscript starting on page 7 as follows:

"To further test our hypothesis, we used published cell-type proportion estimates⁶¹ to determine whether correcting GTEx expression data for the effect of cell-type proportions would result in more homogenous cross-CNS mitochondrial-nuclear correlation profiles. To this end, we included five GTEx regions (see methods) for which we determined the cell-type proportions to be most representative (supplementary figure 6, A and B), and compared the distributions of cross-regional Spearman correlation variances per mitochondrial-nuclear gene pair with and without correction for cell type proportions. Applying this approach, we find that the distributions of variances are significantly different to each other (two-sample Wilcoxon signed rank test, $P < 2.2e-16$), but the medians of both distributions are also significantly higher than 0 (one-sample Wilcoxon signed rank test, $P < 2.2e-16$ for mitochondrial-nuclear distributions derived from both correction strategies) (supplementary figure 6, C and D). As such, we conclude that cell-type proportion is a significant modulator of cross-CNS variation in mitochondrial-nuclear correlations, but regional specialisations still exist after correcting for cell type proportions".

We could only include 5 GTEx regions due to the cell-type data that we had available, and we describe the process of region selection in the methods section of the manuscript starting on page 28 as follows:

"This analysis relied on the assumption that the cell-type proportions represented the cellular composition of the GTEx CNS regions. However, there are several factors that affect

the accuracy of the proportion estimates. Firstly, they were derived using murine brain scRNA-seq data, and so there is a species mismatch here. Additionally, technical factors such as the dissection protocol (excision order and resulting RNA degradation), the size of each target region and the accuracy of tissue excision. Assignment of cell-type proportions spinal cord, which was not dissected from the mice, was observed. Although cell-type proportions are assigned in the dataset, these cannot be considered to accurately represent the celltype composition of the spinal cord. Considering this, we looked to select a subset of regions that were most accurately represented by the cell-type proportion estimates. We calculated 20 axes of variation from GTEx CNS RNAseq data using PEER v1.062 on all samples, obtaining 20 PEER factors for each region. We correlated these with the cell-type proportion estimates to understand whether the cell-type proportions represented the cell-type composition of each GTEx CNS region. The underlying logic being that larger axes of variation would correlate strongly with the cell-type proportion estimates if those estimates are accurate for that tissue.

We identified five CNS regions (anterior cingulate cortex, cortex, frontal cortex, hippocampus, caudate basal ganglia) for which PEER factors explaining most variance in the data were correlated with cell-type proportions. In these regions there were fewer spurious correlations with cell types across the 20 PEER factors, and PEER factors explaining most variance (PEER 1 and 2) were highly correlated ($\rho \geq 0.6$) with cell types, indicating of better representation of these regions by the cell-type proportion data. To obtain a measure for cross-CNS variation of mitochondrial-nuclear relationships, we calculated variances for each mitochondrial-nuclear gene pair across these five regions. We did this for correlation values produced by both correction strategies. Finally, to understand whether the cell-type correction changed cross-CNS variation in mitochondrial-nuclear correlation distributions, we performed a two-sample Wilcoxon signed rank tests with the null hypothesis that the true location shift from standard to standard and cell-type correction-derived distributions was less than zero (aka a negative shift in variance, closer to a median of zero in the latter). We also performed one-sample Wilcoxon tests for each correction strategy to test the null hypothesis that the median of the distribution of variances was equal to zero)".

However, this is a difficult point to resolve completely due to the high cellular complexity of CNS tissue and the lack of comprehensive cell type-specific human data, which means that correction for cell type proportions cannot be performed precisely at present. As a result, we note that some regional variation in correlations is still observed following correction for cell type proportions and as the reviewer suggests this may indicate regional differences in mitochondrial-nuclear gene expression correlations even for the same cell type.

2. The discussion surrounding highly variable nuclear-mitochondrial correlations (starting line 515) focuses on the set of variable genes being anti-correlated with mitochondrial genes, suggesting a mechanism to reduce ROS, while not referring to the variability of these relationships across brain regions implied by the increased variability of the correlations. Significant expansion of the discussion to include the implications of the variability is necessary to frame this result.

Reply: We agree with this statement and have now expanded our commentary on the variable nuclear-mitochondrial correlations to cover not only the directionality of the associations, but with a focussed discussion around why the most variable relationships with mitochondrial gene expression are found to be with nDNA-encoded synaptic marker genes. We have also produced a new supplementary figure (8A) to support the ideas presented therein. We have now included this in the discussion section of the manuscript starting on page 19 as follows:

"The synapse is the site of greatest energy expenditure in the neuron⁶⁵. To match energy supply and demand, the mitochondria in synaptic compartments display structural, biochemical and spatial plasticity⁶⁵. To achieve this necessitates equally flexible maintenance of the mitochondrial proteome, the exact mechanisms of which are not known⁶⁵. Our analysis reveals variable mitochondrial-nuclear relationships being highly significantly enriched for synaptic marker genes, meaning that nuclear-encoded synaptic gene expression and mitochondrial-encoded gene expression are differentially associated across the CNS. We considered the possibility that we may simply be tagging variability in regional mtDNA expression, however, residual TPM values for the 13 mtDNA genes demonstrate consistent cross-CNS expression (supplementary figure 8A), suggesting that this is not a core driver of the regional specificity of mitochondrial-nuclear correlation profiles. It could be that we are indirectly observing mitochondrial plasticity by capturing neuronal subtype-specific variation in nuclear and mitochondrial expression. It is known that neuronal sub-types are energetically specialised⁶⁹, and that CNS tissues have differential neuronal subtype compositions^{67,68}. Sub-cell-type-specific expression modulation as a mechanism to manage local energy requirements at synapses is supported by work finding that heterogenous energy requirements across CNS regions and cell types may necessitate bespoke mitochondrial proteomes⁶⁵. Further to this, molecular evidence shows that several nuclear-encoded mitochondrial genes involved in processes key to mitochondrial plasticity (mitochondrial transcription, fission and trafficking) have been found to exhibit distinct patterns of expression in the neuronal subtypes⁶⁶. Recent work using engineered MitoTag mice coupled with an isolation approach to profile tagged mitochondria from defined cell types, has demonstrated profound cell-type-specific mitochondrial biology serving homeostatic needs to preserve essential functions in cells⁶⁴. And yet, without directional information and cell-type or sub-cell-type specific data, it is difficult to make a firm assertion as to whether the underlying mechanism is anterograde modulation of the mitochondrial genome from the nucleus, or retrograde modulation of the nuclear genome by the mitochondria, or perhaps a feedback loop involving both."

To aid clarity, and to show precision, intersection and P-values for each enrichment, we have edited figure 3. It now represents enrichments as dot plots, better showing pathways that are enriched across gene sets, and highlighting the stark and unique synaptic enrichment of the high variance negative gene set.

3. The heatmaps shown throughout the paper (fig 2, fig 4c, fig 5a) omit colours for non-significant results, but in testing for significance there is no mention of corrections for testing multiple gene lists or brain regions. Some results shown in these heatmaps are left as significant without consideration for the number of tests performed leaving the impression of significant results that could have arisen by chance – though the key findings mentioned in

the text are found at a significance level which accounts for this possibility, except for results on page 16. The authors should make clear which results are robust to multiple testing corrections, and whether this alters the conclusions. This is particularly important in the discussion of neurological disease gene sets, as two of the 5 relevant gene sets fail to produce any results robust to these corrections.

Reply: In the main text all conclusions drawn are based on P values that have been corrected for multiple tests. We apologise if this was unclear, and have now added the correction method used after stating each P value. All of our major conclusions remain unchanged.

We apologise if p-value labels are not clear in figures 2, 4c and 5a, and have now updated each legend to ensure that labelling is fully transparent. For figures 2 and 4c, the colours relate to the s.d. from the mean, whereas in figure 5a the colours represent raw p-values. In all cases we have overlaid asterisks on each box to denote the level of significance that each test passes, with each level clearly defined in the legend as one of the following:

1. * $0.05 / (\text{no. of CNS regions}) < P < 0.05$;
2. ** $0.05 / (\text{no. of CNS regions}) * (\text{no. of gene sets, cell types etc.}) < P < 0.05 / (\text{no. of CNS regions})$;
3. *** $P < 0.05 / (\text{no. of CNS regions}) * (\text{no. of gene sets, cell types etc.})$.

In this way, readers of the article can fully assess the strength of the effect in each case.

For the analysis of neurological disease gene sets, we note that the P values derived from the AD GWAS set and the PD GWAS set are only nominally significant and do not pass multiple testing correction. This is perhaps expected, since these genes are more weakly associated with disease in each case as they were derived from common variant studies. Genes in the “PanelApp adult onset neurodegeneration” set are highly significant after multiple testing correction, as are those in our extended analysis sets considering “early onset dementia”, and “PD and complex PD” PanelApp gene sets. This is now clearly described in the manuscript (see our answer to the first question above).

Minor Comments:

4. A supplementary figure showing the agreement between the nuclear-mitochondrial correlation between the 13 mitochondrial genes would provide support to the aggregation of mitochondrial genes when finding the cross-CNS variance of genes and difference in correlation of genes between AD and control.

Reply: We agreed that this would be a pertinent supplementary figure to support this decision. Thus, we have generated supplementary figure 8B, a plot to show the distribution of nuclear-mitochondrial correlation variances for each nDNA gene across mtDNA genes, split by region. This plot shows that the distribution of variances is highly skewed towards 0 in all CNS regions, indicating that there is very little variance when looking at correlation values within nDNA genes and across mtDNA genes. This lends support to the decision to aggregate across mtDNA genes in the variance analysis.

5. Though mentioned in the methods it is unclear in the main text that the median nuclear-mitochondrial variance was used to curate high and low variance gene lists (pages 10-12)

Reply: We have now made this clear in the results section of the manuscript, starting on page 8 as follows:

"To this end, we calculated the variance of spearman correlation coefficients of each nuclear-mitochondrial gene pair across the 12 GTEx CNS regions, and assigned correlation directionality to each pair (see example in fig. 3B). To reduce redundancy, aggregation of mitochondrial genes was performed, taking the median cross-CNS variance of 13 mitochondrial genes as the representative value for each nuclear gene (supplementary figure 8B). Using this methodology, four gene sets were defined:"

6. When significant is indicated by asterisks (fig 2, fig 4, fig 5) the figure captions have significance marked the wrong way round (eg. * $0.05 < p < 0.05/12$, ** $0.05/12 < p < 0.05/12*6$, *** $p < 0.05/12*6$  * $0.05/12 < p < 0.05$, ** $0.05/12*6 < p < 0.05/12$, *** $p < 0.05/12*6$)

Reply: This has been changed wherever it occurs to:

"* $0.05/12 < P < 0.05$; ** $0.05/12*6 < P < 0.05/12$; * $P < 0.05/12*6$."**

7. Line 364 references a supplementary figure 5b which is not present in the review copy.

Reply: We apologise for this oversight. Supplementary figure 5b has now been included in the supplementary figures file.

REVIEWERS' COMMENTS:

Reviewer #2 (Remarks to the Author):

The authors have addressed all concerns raised, and I have no further concerns. Thank you.